# Improvement of the Similarity Spectral Unmixing Approach for Multiplexed Two-Photon Imaging by Linear Dimension Reduction of the Mixing Matrix

**DOI:** 10.3390/ijms22116046

**Published:** 2021-06-03

**Authors:** Asylkhan Rakhymzhan, Andreas Acs, Anja E. Hauser, Thomas H. Winkler, Raluca A. Niesner

**Affiliations:** 1Biophysical Analytics, Deutsches Rheuma-Forschungszentrum, Berlin, a Leibniz Institute, 10117 Berlin, Germany; 2Nicolaus-Fiebiger-Zentrum, Department of Biology, Division of Genetics, Friedrich-Alexander-University of Erlangen-Nürnberg, 91054 Erlangen, Germany; andreas.acs@fau.de (A.A.); thomas.winkler@fau.de (T.H.W.); 3Charité—Universitätsmedizin Berlin, Corporate Member of Freie Universität Berlin and Humboldt-Universität zu Berlin, Department of Rheumatology and Clinical Immunology, Charitéplatz 1, 10117 Berlin, Germany; anja.hauser-hankeln@charite.de; 4Immune Dynamics, Deutsches Rheuma-Forschungszentrum, Berlin, a Leibniz Institute, 10117 Berlin, Germany; 5Dynamic and Functional in vivo Imaging, Veterinary Medicine, Freie Universität, 14163 Berlin, Germany

**Keywords:** two-photon laser scanning microscopy, multiplexed imaging, fluorescent proteins, similarity spectral unmixing approaches, linear matrix dimension reduction

## Abstract

Two-photon microscopy enables monitoring cellular dynamics and communication in complex systems, within a genuine environment, such as living tissues and, even, living organisms. Particularly, its application to understand cellular interactions in the immune system has brought unique insights into pathophysiologic processes in vivo. Simultaneous multiplexed imaging is required to understand the dynamic orchestration of the multiple cellular and non-cellular tissue compartments defining immune responses. Here, we present an improvement of our previously developed method, which allowed us to achieve multiplexed dynamic intravital two-photon imaging, by using a synergistic strategy. This strategy combines a spectrally broad range of fluorophore emissions, a wave-mixing concept for simultaneous excitation of all targeted fluorophores, and an unmixing algorithm based on the calculation of spectral similarities with previously measured fluorophore fingerprints. The improvement of the similarity spectral unmixing algorithm here described is based on dimensionality reduction of the mixing matrix. We demonstrate its superior performance in the correct pixel-based assignment of probes to tissue compartments labeled by single fluorophores with similar spectral fingerprints, as compared to the full-dimensional similarity spectral unmixing approach.

## 1. Introduction

The orchestration of physiological and pathological phenomena defining health or disease in living organisms involve a highly complex organization of the immune system, in space and time. In order to understand how these phenomena are synchronized, both multi-modal and multiplexed imaging techniques are required. Especially, in the investigation of immune responses, multiplexed—fluorescence or non-fluorescence-based—techniques play a central role [1]. In line with this, studies performed on fixed (static) samples are based on repeated analysis of the same sample, under exactly the same conditions. In recent years, several fluorescence-based technologies [2], such as multi-epitope-ligand cartography—MELC—[3,4] and other cyclic immunofluorescence technologies or co-detection by indexing—CODEX [5]—enabled reliable imaging of up to 100 markers within one sample. Similar results can be achieved by Imaging Mass Cytometry in a shorter time but without sample preservation [6]. In addition, several multiplexed immunohistochemistry approaches have been developed and successfully applied, as reviewed [2].

However, such approaches are not suitable for intact, live tissue, which is highly dynamic on both the cellular and the molecular level. Due to its optically non-linear nature and the implications for optically scattering environment, two-photon microscopy is the method of choice for imaging living organisms, deep within tissue, in a non-invasive or minimally invasive manner. It also allows the simultaneous visualization of several—typically up to four—different cellular or non-cellular tissue compartments over time [1,7,8]. In order to account for the cellular and tissue complexity in living organisms in a dynamic manner, further development of the state-of-the-art technology was necessary, i.e., optimizing the excitation strategy, expanding the fluorescence range towards the near infrared (NIR), and improving spectral unmixing algorithms [7]. 

Extending the spectral range of fluorophore emission to the near-infrared (NIR) region not only reduces autofluorescence, absorption of water and hemoglobin, as well as light scattering in tissue, but also increases the number of available fluorophores, making multiplexed deep tissue imaging feasible [9,10]. Several approaches to achieve two-photon excitation of a broad range of fluorophores within a sample have been developed and applied: sequential single excitation [11,12] by a short-pulsed laser (typically Titanium:Sapphire laser, Ti:Sa), dual excitation [13] by two colocalized short-pulsed lasers (Ti:Sa and Optical Parametric Oscillator, OPO), triple excitation using wavelength mixing of two lasers, i.e., Ti:Sa and OPO, leading to an additional two-color–two-photon excitation [7,14], as well as excitation with an electromagnetic wave with broad continuous spectrum, e.g., super-continuum lasers based on photonic crystal fibers [15]. Among these strategies, the triple excitation with two laser beams synchronized in time and space ensures effective, specific, and simultaneous excitation of all fluorophores and signals of interest, as we demonstrated by intravital imaging of germinal centers [7].

As far as the fluorescence signal discrimination algorithms are concerned, linear spectral unmixing algorithms are typically employed, which require the accurate knowledge of the emission spectra of the single fluorophores and that the number of detectors equals the number of signals to be distinguished. Blind unmixing algorithms [16,17] are able to perform unmixing of up to 15 different sequentially acquired signals without exact previous knowledge of emission spectra, e.g., PICASSO—an approach based on mutual information minimization—but still require acquisition on the same number of detection channels as the number of signals to be resolved. On the other hand, similarity approaches, which rely on reference spectra, are predicted to allow a superior signal assignment quality and to be able to distinguish more signals than the available detection channels [18]. We previously developed a similarity unmixing algorithm (SIMI approach) and applied it to resolve between up to eight signals in multiplexed intravital two-photon microscopy of germinal centers in lymph nodes of mice [7]. Thereby, we used only six detection channels. Still, when the signal is too low or the spectral signatures of the simultaneously excited chromophores are too similar, the SIMI approach is limited in delivering appropriate results with respect to correct signal assignment.

In this work, we used our multiplexed dynamic two-photon excitation imaging method [7] and improved the previously developed similarity unmixing algorithm by applying linear dimension reduction based on numerical mixing matrix diagonalization, as it resulted from the corresponding experimental data. Dimension reduction of diagonalized matrices is a widely used algebraic method [19]; however, its numerical application in similarity fluorescence signal unmixing approaches has not been performed yet. This improvement allowed us to correctly assign the probe in up to 90% of the pixels within segmented single-labeled objects, when simultaneously six signals with partially similar spectral signatures, labelling different cellular and tissue compartments, were detected in murine popliteal lymph nodes, and only four PMT detectors were used. In contrast, the original SIMI algorithm was able, for the same case, to correctly assign the probe in only 25% of the pixels. 

The here described method is a further improvement of the versatile tool we described before [7] to monitor complex dynamic processes in vivo, applicable to the investigation of any organ, in which the motion patterns and communication of various cell populations define tissue function.

## 2. Results

### 2.1. Dimensionality Reduction of the Spectral Mixing Matrix

Similarity unmixing (SIMI) is a numerical pixel-based algorithm, which allows for mixed colors separation based on similarities between overlapping fluorophores and fingerprints (spectral signatures) of the individual fluorophores [7]. The SIMI algorithm originates from the linear unmixing method with the assumption that the total signal S detected on every photo multiplier tube (PMT) channel is linearly proportional to the combination of contributing fluorophores, F_j_ [18]:(1)$$S_{i}=a_{i1}\times F_{1}+a_{i2}\times F_{2}+\ldots =\sum_{j=1}^{m}a_{\mathrm{ij}}F_{j,\;}\;i=1÷n,$$
where i is the detection channel index, j is the fluorophore index, n is the number of detection channels, and m is the number of fluorophores.

The system of linear Equations (1) has the following matrix form:(2)$$\left(\begin{array}{c} S_{1}\\ \vdots \\ S_{n} \end{array}\right)=\left[\begin{array}{ccc} a_{11} & \ldots & a_{1m}\\ \vdots & \ddots & \vdots \\ a_{n1} & \ldots & a_{\mathrm{nm}} \end{array}\right]\times \left(\begin{array}{c} F_{1}\\ \vdots \\ F_{m} \end{array}\right),$$
where $a_{\mathrm{nm}}$ are the elements of the mixing matrix. The matrix Equation (2) can be also written as [19]: (3)$$\left(\begin{array}{c} S_{1}\\ \vdots \\ S_{n} \end{array}\right)=F_{1}\cdot \left(\begin{array}{c} a_{11}\\ \vdots \\ a_{n1} \end{array}\right)+F_{2}\cdot \left(\begin{array}{c} a_{12}\\ \vdots \\ a_{n2} \end{array}\right)+\ldots +F_{m}\cdot \left(\begin{array}{c} a_{1m}\\ \vdots \\ a_{\mathrm{nm}} \end{array}\right)=\sum_{j}F_{j}\cdot \left(\begin{array}{c} a_{1j}\\ \vdots \\ a_{\mathrm{nj}} \end{array}\right),$$
where each column $\left(a_{1j}\ldots a_{\mathrm{nj}}\right)$ represents a fingerprint of fluorophore F_j_. The components of the fingerprint $a_{1j}\;$can be determined in the single-color condition for each given fluorophore ($F_{j}\neq 0,F_{k\neq j}=0,\;$for k=1÷m). Thus, Equation (3) becomes
(4)$$\left(\begin{array}{c} S_{1}\\ \vdots \\ S_{n} \end{array}\right)=S_{\max}\;\cdot \left(\begin{array}{c} c_{1}\\ \vdots \\ c_{n} \end{array}\right)=F_{\max}{}_{j}\cdot \left(\begin{array}{c} a_{1j}\\ \vdots \\ a_{\mathrm{nj}} \end{array}\right),$$
where $c_{1}\ldots c_{n}$ are normalized components of the detection channels measured at the single-color condition, and S_max_ and $F_{\max}{}_{j}$ are constant values. 

In the case of one-probe–one-color condition, the unknown probe can be assigned as a defined fluorophore from the set of known fingerprints based on the following similarity analysis:(5)$$\left(\begin{array}{c} S_{1}\\ \vdots \\ S_{n} \end{array}\right)=S_{\max}\;\cdot \left(\begin{array}{c} b_{1}\\ \vdots \\ b_{n} \end{array}\right){\;\mathrm{matching}\;\mathrm{to}\;F}_{1}\cdot \left(\begin{array}{c} a_{11}\\ \vdots \\ a_{n1} \end{array}\right),{\;F}_{2}\cdot \left(\begin{array}{c} a_{12}\\ \vdots \\ a_{n2} \end{array}\right),\ldots ,F_{m}\cdot \left(\begin{array}{c} a_{1m}\\ \vdots \\ a_{\mathrm{nm}} \end{array}\right).$$

The matching procedure is performed by a fitting approach that minimizes the function$\;f\left(F_{j}\right)=\sqrt{\underset{i}{\sum }{\left(b_{i}-a_{\mathrm{ij}}\right)}^{2}}$, 0<j<m, for each pixel of the measured image. The SIMI method features a unique property, being able to resolve crosstalk issues even in the under-determined condition, i.e., when the number of detection channels are lower than the number of overlapping fluorophores [7]. 

In multicolor systems with more than four probes, their spectral signatures are often distributed not over all detection channels. In such sparse condition, the non-diagonal block components, C and D, of the mixing matrix are close to zero. Thus, the mixing matrix can be transformed as a direct sum of non-zero diagonal block matrices, *A* and *B*: (6)$$\left[\begin{array}{cc} A & C\\ D & B \end{array}\right]={\left[\begin{array}{cc} A & 0\\ 0 & B \end{array}\right]}_{\begin{array}{c} C=0\\ D=0 \end{array}}=A⊕B$$

This mixing matrix transformation allows to reduce the multiplexing system dimensionality and to simplify the similarity analysis. According to the transformation (6), the matrix Equation (2) will be split into a sum of sub-systems with lower dimension
(7)$$\;\vec{S}=\left[\begin{array}{cc} A & 0\\ 0 & B \end{array}\right]\times \;\vec{F}=A\times \;\vec{F}⊕B\times \;\vec{F}$$

The idea of the dimension reduction SIMI (drSIMI) approach is to split the multicolor system into simple subgroups, in accordance with the experimental data, to perform the similarity unmixing of these independent subgroups separately and to combine the processed parts into the complete unmixed image. 

### 2.2. Benchmarking of drSIMI in Standardized Multiplexed Two-Photon Microscopy Data of Single-Label HEK Cells

We imaged single-labeled HEK cells transfected with cyan fluorescent protein (CFP), enhanced green fluorescent protein (eGFP), a monomeric orange fluorescent protein (mOrange2), or a monomeric red fluorescent protein (mKate2), both separately and in mixture. Here, we compared the unmixing performance of the conventional similarity (cSIMI) method, where the entire mixing matrix is used (Equation (2)), with the dimension reduction approach drSIMI (Figure 1c). We used the single-chromophore data to generate two-photon excitation spectra of the fluorescent proteins as well as to retrieve the spectral signature of each chromophore for both similarity unmixing approaches (Figure 1a,b). The probes mOrange2 and mKate2 have a moderate spectral overlap that results in highly distinguishable fingerprints, making their unmixing relatively simple. In contrast, the probes CFP and GFP have strongly overlapping spectra, where their maximum fingerprint components are related to the same detection channel, making color separation more challenging. Therefore, we specifically focused on CFP and GFP crosstalk, for which case the benefits of drSIMI over cSIMI are becoming obvious. Hence, we show the unmixed image layers only for CFP and GFP in Figure 1c. 

For cSIMI, we used the full 4 × 4 mixing matrix in order to perform similarity matching (Equation (5)). The processed fluorophore images are depicted in the second row of Figure 1c. According to the sparse spectral distribution of these fluorophores, the fingerprint components were negligible in the detection channels, in which there was a small or no emission at all, e.g., CFP and GFP in channels 562 and 605, as well as mOrange2 and mKate2 in channels 466 and 525. Under such conditions, we could apply the concept of the drSIMI approach and transform the multiplex system of four fluorophores (CFP, GFP, mOrange2, and mKate 2) into the sum of two sub-systems with two fluorophores (CFP × GFP + mOrange2 × mKate2, Equation (7)). The resulting unmixed images are shown in the third raw of Figure 1c.

In the unmixed images processed with cSIMI, there was still a crosstalk pattern of the CFP probes into the GFP image layer (dashed lines in Figure 1c). In contrast, drSIMI retrieved color separation of these two fluorophores more reliably, i.e., in most of the pixels building up the image of a cell. In order to quantify the unmixing performance of the cSIMI and drSIMI approaches, we analyzed the crosstalk pattern from the CFP probes, which was incorrectly assigned to the GFP layer. For this, we calculated a false assignment rate. This is the ratio of the number of falsely assigned pixels to the total pixel number within a segmented cell, i.e., an object. In our case, the false assignment rate was calculated for segmented CFP-expressing cells having an intensity above background and being assigned to the GFP image layer. We determined false assignment rates of 0.3 for cSIMI (30%) and below 0.03 for drSIMI (3%) (Figure 1d), resulting in an increase of correctly assigned pixels per object, i.e., integrity of the objects, from 70% for cSIMI to above 97% for drSIMI. Since the GFP probe had a minimum spill-over in the 466 nm channel, the false assignment rate of the GFP probe into the CFP image layer was negligible.

### 2.3. Superior Performance of drSIMI for Resolving Signals in Multiplexed Deep-Tissue Two-Photon Microscopy Data

In Figure 2a, the fingerprints of fluorophores and a second harmonic generation (SHG) of collagen on the surface of a lymph node capsule are presented in normalized form. These fingerprints were determined in situ from the specific features of the given fluorescence probe. CFP labels the cell membrane, eGFP is expressed in the nuclei, yellow fluorescent protein (YFP) and a variant of red fluorescent protein isolated from *Discosoma* (DsRed) are contained in the cytoplasm [20]. The fingerprint of macrophages, displaying a high degree of endogeneous fluorescence, has been defined from a signal in their distinctive location in lymph nodes. The fingerprint of SHG has been also defined from the specific tissue morphology. The values of all components in the mixing matrix are presented in Figure 2b. The non-diagonal blocks shown in red boxes are close to zero, implying that the detection channels 466 and 525 contained minimal or no signal of the probes DsRed, SHG, and macrophages. The same situation occurred with channels 595 and 655, which showed minimal signals of the CFP, eGFP, and YFP probes. Mathematically, the full system with four channels and six probes can be transformed into two independent subsystems of two channels and three corresponding probes (Figure 2c). Therefore, we could apply the drSIMI approach on this palette of fluorophores and compare its outcome with those of the conventional similarity (cSIMI) method.

In Figure 3, a comparison of the unmixing performance between the cSIMI method and the drSIMI approach is shown. In the case of cSIMI, the matrix equation with the full mixing matrix was applied on four channels and six probe compartments as one complete system. For the drSIMI approach, a different strategy was performed, as the full mixing matrix could be diagonalized and split into two independent parts, where two sets of channels were connected to three corresponding probes. Two mixing matrices for these independent subsystems were separately measured in situ (Figure 2c). The raw images were separated into two sets of data, and each of them was processed according to the corresponding connection, the channels 466 and 525 with the probes CFP, eGFP, and YFP, and the channels 595 and 655 with the probes DsRed, SHG, and macrophages. After post-processing, the unmixed results were merged into one multicolor image (Figure 3a). In order to demonstrate the unmixing performance, images at a higher magnification are shown in Figure 3b. A poor unmixing capacity can be seen for CFP, eGFP, and YFP in the case of the cSIMI method. In Figure 3b, the smaller fields of view clearly show that the cSIMI method incorrectly assigned the majority of pixels stemming from the CFP, eGFP, and YFP probes, whereas the drSIMI approach determined the signal pixels entirely, within the cellular or tissue object, and correctly. The fingerprint elements of the CFP, eGFP, and YFP probes are relatively similar, which makes the unmixing process more difficult, unlike what has been observed for other probes, i.e., DsRed, SHG, and macrophages, with a completely different fingerprint profile. For the probes with different fingerprint profiles, both algorithms demonstrated comparably good unmixing performance and correct assignment of the probe pixels. A quantitative comparison of the unmixing quality was done based on the Pearson correlation analysis between the raw data with unmixed images [21]. The drSIMI approach outperformed the cSIMI method in the case of spectrally close probes and demonstrated the same performance for probes with different spectral signatures (Figure 3c).

In the zoom-ins in Figure 3b, the CFP, GFP, and YFP probes were poorly assigned on a pixel basis by cSIMI, showing incomplete object integrity. In contrast, the drSIMI approach demonstrated less false assignment of the fluorophore probes, which led to much higher object integrity. In order to quantify the correct pixel assignment in live tissue images by cSIMI as compared to drSIMI, we determined the integrity of the probe assignment within a biological compartment on a single-object basis, e.g., on a single cell basis. Thereby, we calculated the integrity assignment rate as the ratio of the number of pixels correctly assigned to the image layer of the respective probe to the total amount of pixels contained in a segmented raw image object. The resulting graph of Figure 3d shows that object integrity in the case of cSIMI unmixing was below 0.25 for all three fluorophores (25% of the pixels within an object were correctly assigned), while the drSIMI approach showed up to 0.9 integrity assignment rate (90% of the pixels within an object were correctly assigned). The segmented objects within raw images have, by definition, integrity assignment rate equal to 1 (100%). This implies that for overlapping probes with close spectral features, the drSIMI approach will show better unmixing and pixel-based probe assignment performance than the conventional method. This conclusion confirms that less components in the fingerprints and more distinct spectral signatures of the overlapping probes result in a better signal separation.

## 3. Discussion

Understanding pathophysiological phenomena requires a thorough investigation in their original environment, i.e., the living tissue, in a spatio-temporal manner. Two-photon microscopy and, more recently, three-photon microscopy [22,23] have proven to be the most versatile tools fulfilling these requirements [24]. They brought new insights into cellular dynamics, communication, and tissue functions in different areas such as neurosciences [25], immunology [26], and oncology [27]. Since most pathophysiological phenomena, independent of the organs they involve, are governed by a complex dynamic interplay of various cellular and non-cellular tissue compartments, simultaneous spectral multiplexing is required in multi-photon imaging of live tissue, in order to elucidate underlying mechanisms. Besides, multi-modal imaging technologies retain the potential to translate in-depth mechanistic understanding of disease pathogenesis into clinical settings, for example, by combining multi-photon microscopy with imaging technologies used in diagnostics, such as magnetic resonance imaging, positron emission tomography, or optical coherence tomography [28].

We previously developed an intravital multiplexed two-photon microscopy approach based on a synergistic strategy, combing effective triple excitation of a broad spectral range of fluorophores with a non-deterministic spectral unmixing algorithm (SIMI approach) and demonstrated its performance for dynamically imaging germinal center dynamics [7]. However, despite the high performance and flexibility of the SIMI approach, it delivered poor results when the spectral signatures of the used fluorophores were too similar. Specifically, only a small fraction of pixels within a segmented object (typically, a cell) was correctly attributed to the signals of CFP, eGFP, and YFP by cSIMI, when only two detection channels were available.

Here, we propose to reduce the dimensionality of the mixing matrix, which brings together the signatures of the detection channels and those of the fluorophores used for labelling. Thereby, we take advantage of the fact that the matrix is quasi-diagonal and allows a straightforward clustering of detection and spectral signatures. Compared to the cSIMI algorithm, the newly developed approach, drSIMI, showed a superior unmixing performance in the case of the similar spectral signatures of CFP and eGFP expressed in HEK cells and of CFP, eGFP, and YFP probes in murine lymph nodes. Specifically, the false pixel-based assignment of the CFP signal in the eGFP image layer, in HEK cells, was reduced from 30% for cSIMI to under 3% for drSIMI, leading to an increase of the integrity assignment rate, i.e., the fraction of pixels correctly assigned to the correct signal layer, from 70% to 97%. In murine lymph nodes, the integrity assignment rate could be increased from 25% for cSIMI to 90% for drSIMI in the case of CFP-, eGFP-, and YFP-expressing B cells. When the fingerprints of the signals were far apart, as in the case of mOrange2 and mKate2 or DsRed, SHG, and macrophages, the performance of the two algorithms was comparably good.

A remaining challenge for the here presented similarity approach as well as for most currently available spectral unmixing algorithms for multiplexed fluorescence imaging data is posed by the situation in which single tissue compartments are labelled by multiple probes/fluorophores, which leads to a more complex mixing matrix. Prospectively, further efforts to achieve quasi-diagonalization of such complex mixing matrices or transformations towards sparsely occupied matrices are required in response to this challenge.

Still, the here presented improvement of the non-deterministic spectral unmixing approach based on the similarity of the acquired signal with single-species signatures is generally applicable to other two- or three-photon microscopy data, providing major advantages over existing algorithms such as the state-of-the-art linear unmixing approach. These advantages are (i) high flexibility, since the number of fluorescence and non-fluorescence signals that can be resolved is higher than the number of detection channels and (ii) possibility to resolve several signals with very similar spectral signatures, if proper numerical quasi-diagonalization of the mixing matrix is achieved by an appropriate choice of the spectral location of the detection channels.

## 4. Material and Methods

### 4.1. Multiplexed Two-Photon Laser-Scanning Microscope Setup

Multiplexed two-photon laser-scanning fluorescence imaging experiments were performed as previously described [7], using a specialized laser-scanning microscope based on a commercial scan head (TriMScope II, LaVision BioTec, Bielefeld, Germany). A near-infrared laser (Ti:Sa, Chameleon Ultra II, Coherent, Dieburg, Germany) and an infrared laser (OPO, APE, Berlin, Germany) were employed for sample excitation. The Ti:Sa and OPO beams, both linearly polarized, were superimposed in the scan head using a dichroic mirror (T1045, Chroma, Bellows Falls, VT, USA). A water-immersion objective lens (20×, NA 1.0, Plan-Apochromat, Carl Zeiss, Jena, Germany) was used to focus the co-localized laser beams into the sample. The laser pulse trains were temporally synchronized using a piezo-motorized delay stage (MS30, Qioptiq, Göttingen, Germany), as the relative divergence of the two lasers was controlled by beam expanders. The power of both lasers was independently controlled by combinations of λ/2 wave-plates and polarizers. The ultrashort pulses of both lasers were compressed using external compressors. Fluorescence, second-harmonic generation (SHG), and wavelength mixing signals were collected in the backward direction using a dichroic mirror (775, Chroma, Bellows Falls, VT, USA) and directed to four photo multiplier tubes (H7422, Hamamatsu, Japan). All PMTs were assembled in a detection system with different optical channels, where every channel was determined by an individual fluorescence filter and a set of dichroic mirrors, as indicated in the manuscript: 466 ± 20 nm, 525 ± 25 nm, 562± 20 nm and 605± 20 nm for in vitro experiment; 466 ± 20 nm, 525 ± 25 nm, 595 ± 20 nm, and 655 ± 20 nm for live tissue experiment. In all imaging experiments, we used an average maximum laser power of 10 mW to avoid photodamage. The acquisition time for an image with a field of view of 500 µm × 500 µm and a digital resolution of 1024 × 1024 pixel was 944 ms. We acquired 40 µm z-stacks (z-step 2 µm).

### 4.2. Data Analysis

The cSIMI algorithm was integrated as PlugIn in the linear unmixing PlugIn of Fiji/ImageJ (version 1.3, accessed in 24 February 2015) written by Joachim Walter. The custom-written code is available from the authors upon request. The Pearson correlation coefficients were calculated to verify result quality in Fiji/ImageJ using the JACoP PlugIn [21].

### 4.3. HEK Cells Transfection and Imaging

We prepared two types of isolated HEK cell samples. First, we prepared samples containing HEK-293T cells expressing a single color of one out of the four FPs. For each single-labeled fluorophore, we acquired images on all four PMT channels and extracted a fingerprint, also known as a signature, of a given fluorescent protein. The fingerprint represents a ratio of relative intensities in different PMT channels and serves as the main criterion in our spectral unmixing analysis. Second, we prepared samples containing a mixture of single-labeled HEK-293T cells, each expressing one of four FPs. To achieve “one cell–one color” labeling in the sample mixture, first we transfected HEK-293T cells separately with different FP-encoding vectors and then mixed these cells in equal proportions on one collagen-coated plate. We transfected HEK cells following the protocol provided for Lipofectamine 3000 (ThermoFischer Scientific, Waltham, MA, USA), using vectors encoding eCFP, eGFP, mOrange2, and mKate2.

### 4.4. Mice

All mice used were on a C57Bl/6 background. We used F1 mice from a breeding of Rosa26-Brainbow2.1 mice [29] (obtained from the Jackson Laboratory) with Rosa26-Cre^ERT2^ mice [30] (obtained from Taconic Biosciences, Inc., Leverkusen, Germany), as previously described [7]. 

### 4.5. Preparation of Freshly Explanted Popliteal Lymph Node for Imaging

Mice were sacrificed by cervical dislocation. The popliteal lymph node was freshly explanted, fixed on a Petri dish using tissue glue, and immersed in RPMI (Roswell Park Memorial Institute) medium containing 10% FCS (fetal calf serum). The Petri dish was placed on the microscope table for two-photon imaging. A temperature of 37 °C was maintained at all times during the experiment using a heating foil.

### 4.6. Recording of Two-Photon Spectra of Fluorescent Proteins, Second-Harmonic Generation, and Autofluorescence

In order to ensure optimal excitation of the fluorescent proteins, we measured their two-photon excitation spectra in situ, as we previously described [7,13], over a wide wavelength range by using Ti–Sa (760 ≤ λ_Ti:Sa_ ≤ 1040 nm) and OPO (1060 ≤ λ_OPO_ ≤ 1300 nm) for excitation. The raw data were corrected for background signal, peak photon flux, including the squared laser power, photon energy in pulse peak, pulse width, repetition rate of the lasers, and excitation volume at each excitation wavelength.

## Figures and Tables

**Figure 1 ijms-22-06046-f001:**
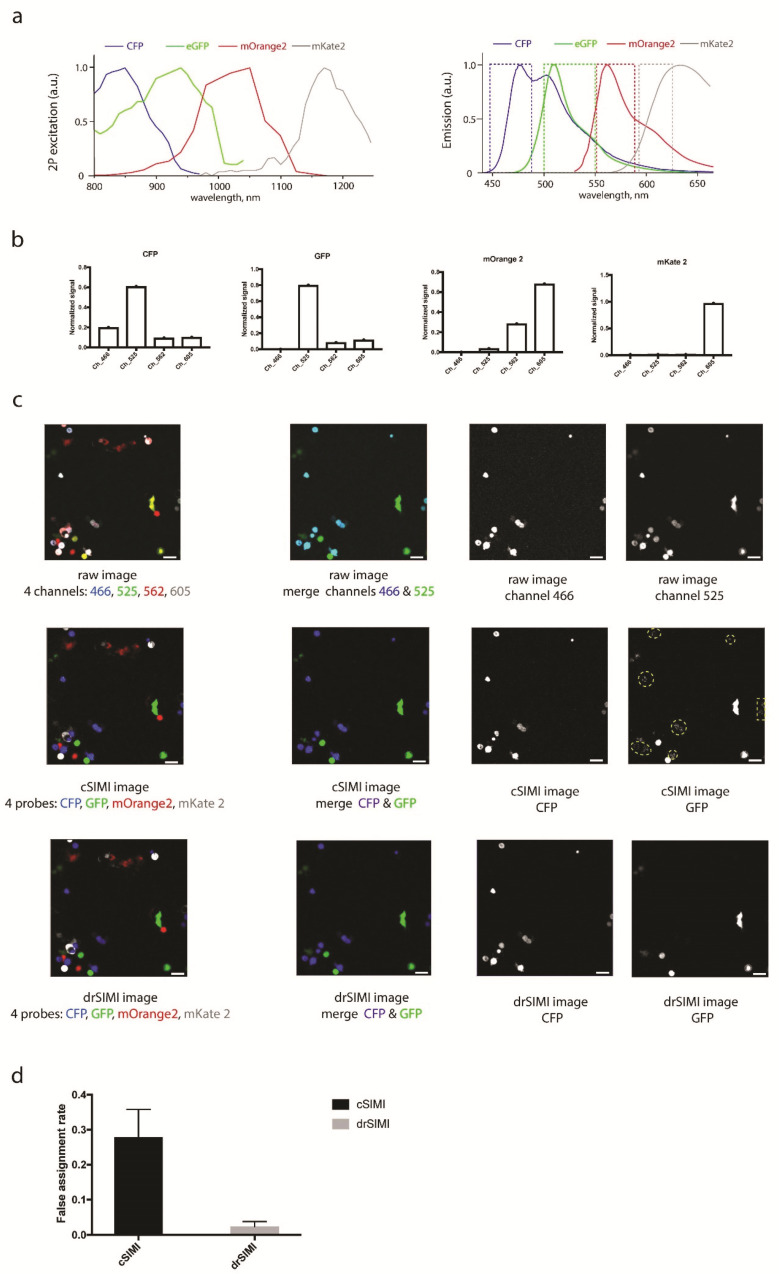
(**a**) Two-photon excitation and emission spectra of cyan fluorescent protein (CFP), enhanced green fluorescent protein (eGFP), a monomeric orange fluorescent protein (mOrange2), and a monomeric red fluorescent protein (mKate2). (**b**) Spectral signatures (fingerprints) of CFP, eGFP, mOrange2, and mKate2 used for both drSIMI and cSIMI approaches. (**c**) Two-photon fluorescence images of mixed HEK cells transfected with CFP, GFP, mOrange2, or mKate2, together with the results of cSIMI and drSIMI. First column: merged fluorescence images containing all four probes; second column: merged fluorescence images containing only the CFP and eGFP probes, i.e., the probes that represent a spectral unmixing challenge; third column: CFP image layer; fourth column: eGFP image layer. The yellow dashed-line areas depict a crosstalk pattern of the CFP probes into the GFP image layer. (**d**) Quantification of the unmixing accuracy (false assignment rate of the chromophore within a segmented object) of the pixel-based similarity unmixing approaches (0—no false assignment; 1—no correct assignment). Scale bar, 50 μm.

**Figure 2 ijms-22-06046-f002:**
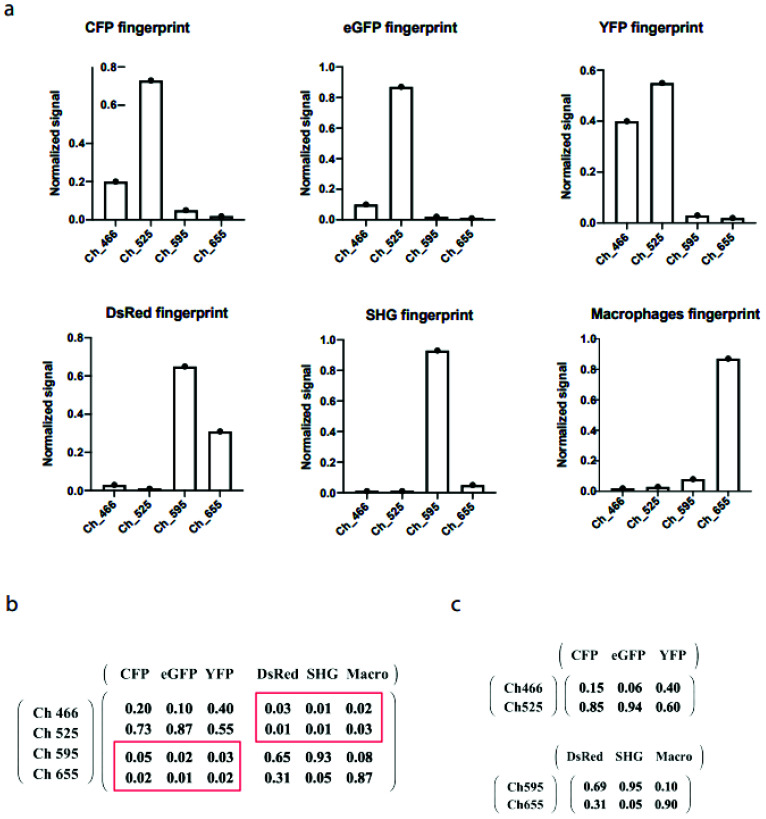
(**a**) Fingerprints of CFP, eGFP, YFP, DsRed, SHG, and macrophages. (**b**) Full mixing matrix for the cSIMI method. (**c**) Two independent mixing matrices for the drSIMI approach.

**Figure 3 ijms-22-06046-f003:**
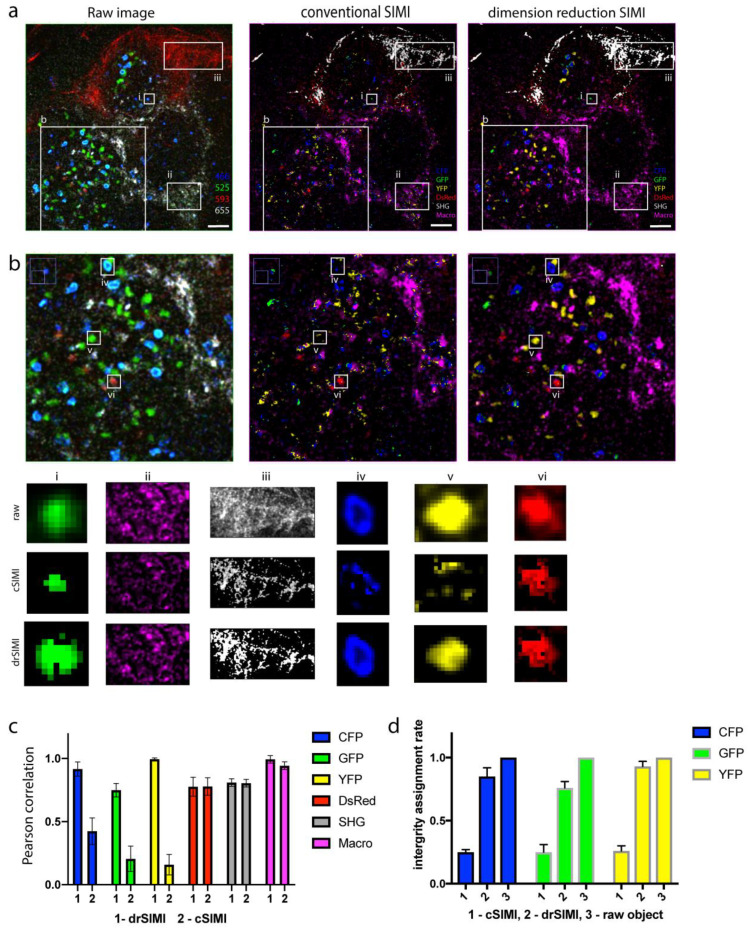
Comparison of unmixing performance between the cSIMI approach and the drSIMI unmixing. (**a**) Raw image, cSIMI unmixed, and drSIMI processed images of a popliteal lymph node from a fluorescent reporter mouse. Color coding: (raw image) blue—PMT channel 466, green—PMT channel 525, red—PMT channel 595, grey—PMT channel 655; (SIMI conventional and block reduction) blue—cyan fluorescent protein (CFP), green—enhanced green fluorescent protein (eGFP), yellow—yellow fluorescent protein (YFP), red—red fluorescent protein isolated from *Discosoma* (DsRed), grey—second harmonics generation (SHG), magenta—macrophages (autofluorescence). (**b**) Zoom-ins from (**a**) show clearly better unmixing performance of the drSIMI approach compared to the cSIMI approach. Field of view of each unmixed compartment, chosen for colocalization analysis. (**c**) Quality control of unmixing performance of drSIMI and cSIMI unmixing based on Pearson correlation coefficient. (**d**) Quantification of the unmixing performance: graph of integrity assignment rates for the cSIMI and drSIMI approaches. The integrity rates are normalized to segmented objects in the raw images. Scale bar, 50 μm.

## Data Availability

All raw data used in this study are available online at: http://doi.org/10.5281/zenodo.4884946, accessed on 10 May 2021.

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
