# Peer review of "Improvement of the Similarity Spectral Unmixing Approach for Multiplexed Two-Photon Imaging by Linear Dimension Reduction of the Mixing Matrix"

_ijms, 2021, doi:10.3390/ijms22116046_

Round 1
Reviewer 1 Report
Authors did a good job in re-writing the main text, in their inclusion of both the theory behind and control experiments to show the feasibility of their method and presented quantifications. The current paper is very readable and the logic of the narrative can be easily followed. I appreciate that the authors not only mention the strengths of the method but also (although very briefly) state its limitations. I believe the current approach could be a nice tool for other people. I now only have a few minor comments:
- Figure axis and labels are hardly readable.
- Figures are of very low quality, please replace those with high quality images.
- I find Figure 1 redundant. Since the concept of the paper is now clearly explained with the math (eqs 2-7), I’d suggest to remove fig. 1.
- Figure 4 a and b panels: Lines connecting small ROI’s with large image are confusing. Would it be possible to rearrange them, perhaps using numbers to identify their origin and remove the lines?
- It would be useful if the code could be included in the ImageJ Plugin, together with some documentation on how to use it. That would strongly encourage its use and distribution in the community.
Author Response
Authors did a good job in re-writing the main text, in their inclusion of both the theory behind and control experiments to show the feasibility of their method and presented quantifications. The current paper is very readable and the logic of the narrative can be easily followed. I appreciate that the authors not only mention the strengths of the method but also (although very briefly) state its limitations. I believe the current approach could be a nice tool for other people. I now only have a few minor comments:
We thank the reviewer for the kind comments and are glad we could respond his major concerns regarding our manuscript.
- Figure axis and labels are hardly readable.
We changed the graphs in the Figures so that the labels can be better read.
- Figures are of very low quality, please replace those with high quality images.
High quality Figures are provided. Compressing the manuscript in the pdf-file led to the poor quality of the images.
- I find Figure 1 redundant. Since the concept of the paper is now clearly explained with the math (eqs 2-7), I’d suggest to remove fig. 1.
We removed Figure 1.
- Figure 4 a and b panels: Lines connecting small ROI’s with large image are confusing. Would it be possible to rearrange them, perhaps using numbers to identify their origin and remove the lines?
We rearranged Figure 4 and excluded connecting lines.
- It would be useful if the code could be included in the ImageJ Plugin, together with some documentation on how to use it. That would strongly encourage its use and distribution in the community.
The code of SIMI algorithm has been already implemented inside the ImageJ PlugIn Spectral unmixing written by Joachim Walter. We will include a thorough documentation in the code and upload it in the open-source platform (ImageJ/FIJI).

Reviewer 2 Report
The manuscript by Rakhymzhan et al. proposes an improvement of a previously published spectral unmixing algorithm for two-photon microscopy. The principle of this improvement consists in neglecting the weakest terms in the mixing matrix.
As a general comment, the reason for neglecting out-of diagonal terms is not mathematically justified or discussed, which cannot be considered as acceptable, although the experimental proof of principle supports this data processing.
In addition, the manuscript remains elusive about whether the proposed algorithm truly unmix spectral channels or can only assign a color channel to isolated probes. According to the proposed mathematical treatment, it seems that the solution is actually a mix of both: True unmixing can be achieved between subsets of spectra but inside these subset, channel attribution can only be achieved if probes are spatially separated.
Author Response
The manuscript by Rakhymzhan et al. proposes an improvement of a previously published spectral unmixing algorithm for two-photon microscopy. The principle of this improvement consists in neglecting the weakest terms in the mixing matrix.
As a general comment, the reason for neglecting out-of diagonal terms is not mathematically justified or discussed, which cannot be considered as acceptable, although the experimental proof of principle supports this data processing.
We need to apologize we didn't emphasize sufficiently that the reason for excluding the weakest terms of the mixing matrix is not of analytical nature but is a numerical approximation, which results from the experimental proof. As far as we verified, the weak terms can be neglected as long as each entity (and each pixel) contains only a single signal. The main benefit of excluding the weakest terms, i.e. reducing the dimensionality of the initial mixing matrix, is a great simplification of the unmixing procedure, which yields to a much more reliable signal attribution on a pixel basis. Moreover, pixels belonging to one entity are all attributed to one and the same signal, thus, ensuring the integrity of the entity (e.g. cell). We clarified this aspect in the revised version of our manuscript.
In addition, the manuscript remains elusive about whether the proposed algorithm truly unmix spectral channels or can only assign a color channel to isolated probes. According to the proposed mathematical treatment, it seems that the solution is actually a mix of both: True unmixing can be achieved between subsets of spectra but inside these subset, channel attribution can only be achieved if probes are spatially separated.
We thank the reviewer for pointing us out that our manuscript may be misleading in this respect. We better stated in the revised version of our manuscript that the here presented improvement is not related to unmixing signals within a pixel or an entity, but is demonstrated for spatially separated signals, i.e. each pixel (and each entity) contains only one single signal – this is the real case in our experiments. In order to deal with entities (and pixels) containing more than one signal, the approach needs to be further developed, this being the subject of future work. We better pointed out these two aspects in the revised manuscript.

This manuscript is a resubmission of an earlier submission. The following is a list of the peer review reports and author responses from that submission.
Round 1
Reviewer 1 Report
Rakhymzhan et al develop a matrix splitting method to overcome the challenge to determine the individual fluorescence components of various fluorophores imaged in an under-determined scenario. The methodology presented could be of potential interest to the community working on intravital imaging. However, in its current form the work lacks scientific rigor. There is no quantification of performance nor statistical evaluation of presented results. All is presented is a single image. Additionally, authors have no metric to estimate accuracy or error.
For a method like this it would be desirable to work from first principles and delineate a theoretical approach to solve the problem. Then, using calculations show the feasibility of the proposed approach. Finally, provide experimental evidence that the method indeed performs as predicted. The current work bypasses the first two points and only anecdotically presents a single image as method validation.
Main points:
- The authors could use theoretical spectra and demonstrate using calculations/simulations the validity of their idea. For example, see https://www.biorxiv.org/content/10.1101/361790v1.full and https://www.biorxiv.org/content/10.1101/2021.01.27.428247v1.full
- Then, authors could provide evidence for their method using controls with binary or ternary mixes of fluorophores, before imaging complex living systems.
- Authors write (p6 L165): ”A poor unmixing capacity can be seen for CFP, eGFP and YFP in the 165 case of the cSIMI method.” Yet, on the figure it is unclear which image represents each channel.
- (p6 L173) “The quantitative comparison of the unmixing quality was done based on the Pearson correlation analysis between the raw data with unmixed images”. This is very confusing. If Pearson correlation is done between the raw data and the unmixed image the correlation will be higher the more the unmixed is similar to the raw, meaning that the perfect unmixed is identical to the raw (leaving the raw untouched and containing all mixed spectra).
- (p6 L175) “The drSIMI approach outperforms the cSIMI method for 175 the case of spectrally close probes and demonstrates the same performance for probes 176 with different spectral signatures (Fig. 3c)“. There is no quantification of performance (how far or close is their method to the ideal case of no spectral overlap?) nor statistics of any kind.
Minor comments.
- The sentences: “Two-photon microscopy is there-48 fore the method of choice, since it allows the simultaneous visualization of several typically up to 4 - different cellular or non-cellular tissue compartments over time” is not correct. 2P is required for imaging deeper into the tissue but not because 1P cannot do 4 (or 5) wavelengths.
Reviewer 2 Report
The proposed article is well written and clearly presents the strategy proposed by the authors to efficiently distinguish single fluorophores with a low number of detectors.
The novelty might be reduced, since it is a further improvement of an algorithm already presented by the same authors, nonetheless it might be useful for the readers to know that this method can increase its reliability.
I would suggest, for sake of honesty, to clearly state at the end of the introduction that the proposed method, as well as the original algorithm, is working only in case of single-species signatures, as mentioned quickly only at the end of the discussion (line 227), to better define and discuss the limitations and the scenarios of its applicability.
Also, a typo at line 61, the Ti in Ti:Sa is Titanium, not Titan